# Green synthesis of zinc oxide nanoparticles using *Catunaregam spinosa* (Thunb.) triveng for biologicals applications

Rabina Baraili[1], Ishwor Pathak[2], Sugam Sharma[3], Manisha Bhusal[1], Khaga Raj Sharma [1]*

1  Central Department of Chemistry, Tribhuvan University, Kirtipur, Kathmandu, Nepal, 2  Department of Chemistry, Amrit Campus, Tribhuvan University, Kathmandu, Nepal, 3  Department of Computer Engineering, Kathmandu Engineering College, Kathmandu, Nepal

* khaga.sharma@cdc.tu.edu.np

## Abstract

Green synthesis, which creates nanoparticles from natural sources such as microorganisms or plant extracts, is a sustainable alternative to traditional chemical methods since it employs less hazardous chemicals and has fewer negative environmental consequences. In this study, the antioxidant, antibacterial, and toxicological properties of ZnO nanoparticles were investigated by synthesizing them using an aqueous extract of *Catunaregam spinosa* (*C. spinosa*) leaves. X-ray Powder Diffraction (XRD), Field Emission Scanning Electron Microscopy (FE-SEM), Energy Dispersive X-ray Analysis (EDX), Fourier Transform Infrared Spectroscopy (FTIR), and UV-vis spectroscopy were used to confirm the formation and properties of ZnO-NPs. FE-SEM investigation indicated a nearly spherical form, and XRD computed an average crystallite size of $12.39 \pm 3.84$ nm. The UV-vis spectrum showed a high absorption peak at 366 nm. FTIR indicated the existence of bioactive functional groups, which are important for nanoparticle capping and stability. ZnO-NPs showed antibacterial efficacy against *Escherichia coli*, *Shigella sonnei*, *Klebsiella pneumoniae*, and *Staphylococcus aureus* at 50 µg/mL. The inhibition zones were 9 mm, 13 mm, 16 mm, and 15 mm, respectively. *Klebsiella pneumoniae* and *Staphylococcus aureus* had the same minimum inhibitory concentration (MIC) of 6.25 mg/mL. Nanoparticles also demonstrated significant antioxidant activity in the DPPH experiment. Toxicity evaluation against *Artemia salina* yielded an $LC_{50}$ of $55.20 \pm 16.19$ µg/mL, indicating dose-dependent effects. Overall, this study found that ZnO-NPs generated with *Catunaregam spinosa* leaf extract have promising antibacterial, antioxidant, and toxic characteristics, indicating potential biological applications.

**Data availability statement:** All relevant data are available at https://doi.org/10.5281/zenodo.16785370.

**Funding:** The author(s) received no specific funding for this work.

**Competing interests:** No conflict of interest on this research paper.

## Introduction

The fundamental goals of the rapidly growing field of nanotechnology are the creation, modification, and application of materials ranging in size from 10 to 500 nm for a variety of medical procedures and drug delivery systems [1]. It is seen as a benefit to modern medicine since it has progressed to the point where new potential for nanoscience exists, particularly in nanodrug, gene transport, and biosensing [2,3]. Because of their surface plasmon resonance, solubility, adhesiveness, and chemical stability, metal and metal oxide (MO) nanoparticles (NPs) are expected to spread easily throughout the ecosystem [4–6]. The most recent studies on antibacterial, antioxidant, and anticancer activities have concentrated on ZnO, CuO, $SnO_2$, $Ag_2O$, $Fe_2O_3$, CaO, NiO, MgO, and AuNPs [7,8].

A wide range of chemical and physical techniques has been used to synthesize NPs, including thermal, sol-gel, hydrothermal, sonochemical, rapid precipitation, and microwave irradiation techniques [9]. Bioresource-based synthesis is a potential solution to the risk of using harmful chemicals in various synthesis processes, as it is less toxic, more ecologically friendly, and more adaptable [10]. Polyols, which are found in different plant parts such as peels, leaves, roots, fruits, seeds, and bark, can stabilize metallic nanoparticles by acting as chelating and capping agents during their rapid production [11,12]. It emphasizes the efficacy of "green synthesis," which reduces synthesis costs and energy needs when compared to chemical or physical NP synthesis [13,14]. Economic interest in developing ecologically friendly ZnO nanoparticles is significant due to their exceptional durability, bioactivity, and biocompatibility [15].

Zinc oxide (ZnO) is a promising biomedical candidate with anti-inflammatory, antibacterial, antioxidant, biosensing, anti-cancer, and cell imaging properties [16,17]. It has also been successfully used to treat diabetes since zinc is known to preserve the structural integrity of insulin [18]. Furthermore, due to their exceptional brightness, ZnO-NPs are among the best options for bioimaging [19–21]. Although interest in ZnO's exceptional antibacterial activity has grown, scientists are still working to figure out how ZnO-NPs develop their therapeutic properties.

*C. spinosa* (mountain pomegranate) is a medium-sized, perennial wild shrub in the *Rubiaceae* family [22]. It is a thorny shrub that may grow up to 4,000 feet above sea level and reach a height of 5 meters. It features oblong, glossy, pubescent leaves with honey-like fragrances from its white blossoms. Indigenous people utilized the plant's stem and bark to heal muscle soreness, and its roots to treat dandruff. Aside from wound healing, *C. spinosa* leaves are used to treat fever, gastrointestinal issues, tumors, piles, snake bites, and diarrhea. Some investigations have found that *C. spinosa* has antihyperglycemic, antimicrobial, hepatoprotective, anti-inflammatory, and anticataleptic effects [23,24].

Although nanoparticles have been manufactured from a variety of plant extracts, very few investigations have been conducted on nanoparticles, particularly ZnO, derived from *C. spinosa*. There are very limited instances of green NP synthesis using *C. spinosa* fruits and leaves that have shown effective, eco-friendly, and unique medicinal properties, paving the way for the development of cutting-edge

nanotechnology solutions [25,26]. To summarize, this work emphasizes the plant's high phytochemical content for eco-friendly synthesis, which is defined by optimal synthesis, full characterization, and evaluation of antibacterial, antioxidant, and toxic capabilities.

*C. spinosa* leaves were chosen because they contain bioactive secondary metabolites and are readily available and regenerative, making them a viable and sustainable choice for green synthesis. Following optimization, ZnO-NPs were characterized by FTIR, EDX, FE-SEM, UV-Vis, and XRD studies. Furthermore, the plant's role as a natural stabilizer and reducer in the synthesis of metal oxide nanoparticles was explored, and the ZnO-NPs' cytotoxic, antibacterial, and anti-oxidant capabilities were assessed. The use of *C. spinosa* extract, which has received little attention for the production of green nanoparticles, in conjunction with a thorough assessment of their biological activity, is the most creative and powerful aspect of this work. Fig 1 shows some of the bio-medicinal applications of ZnO-NPs.

## Materials and methods

### Collection of plant material and extraction of metabolites

The healthy and fresh leaves of *C. spinosa* were collected in June 2023 from a community forest in Bardiya district (mid-western Nepal) at latitude 28° 9′ 8″ N and longitude 81° 29′ 2″ E. It's 479 feet above sea level. The herbarium was recognized by the National Herbarium and Plant Laboratories in Godawari, Lalitpur, Nepal, and with the voucher code RB-002 (KATH). The specimen was stored in the herbarium at a temperature of 15–16 °C in a standard cabinet. Fig 2 shows an image of *Catunaregam spinosa* (Thunb.) Triveng. The leaves were washed, dried in the shade, and then pulverized into a fine powder. Next, 100 milliliters of deionized water and five grams of powder were put in glasses and spun for half an hour at 60° C. After filtering the mixture, the extract was refrigerated for later use. Because the sample is immersed in the

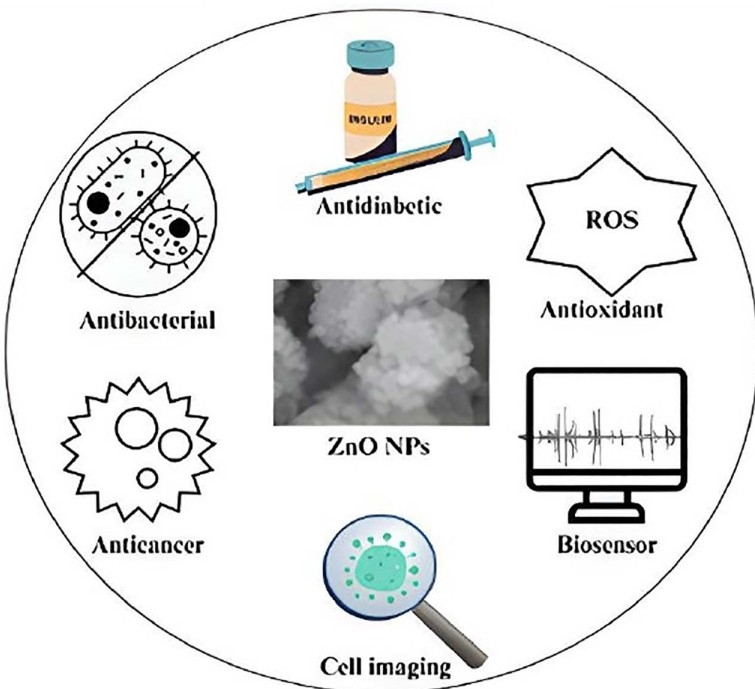

**Fig 1. Bio-medicinal applications of ZnO-NPs.**

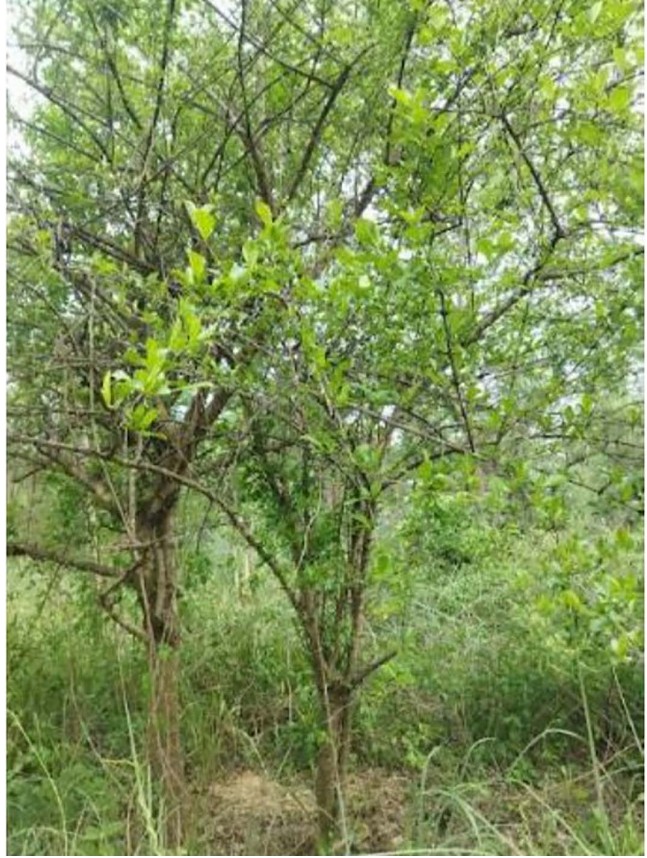

**Fig 2. *Catunaregam spinosa* (Thunb.) Triveng.**

solvent for a long time during the extraction process, a small amount of non-polar component may be extracted despite the solvent's polarity, potentially aiding in the formation of nanoparticles.

## Synthesis of ZnO-NPs

After preparing 500 mL of a 0.13 M zinc nitrate hexahydrate solution, 100 mL of the extract was mixed with 5 mL of zinc nitrate hexahydrate to produce ZnO-NPs. This mixture was then agitated for an hour at 60 °C. After centrifugation at 9000 rpm, the mixtures were calcined at 400 °C for two hours. Fig 3 depicts the schematic route for the synthesis of ZnO-NPs.

Fig 4 shows a proposed reaction pathway for the formation of ZnO-NPs using *C. spinosa* leaf extract, in which the zinc precursor and the extract's functional components ligate. On the word of the literature evaluations, organic chemical groups such as phenolics and flavonoids found in plant extracts operate as ligands. These components, which contain hydroxy aromatic ring groups, interact with zinc ions to form complex ligands. Nanoparticles are created and stabilized through nucleation and shaping processes. When the organic mixture is heated at 400 °C, it directly breaks down and produces ZnO nanoparticles [27].

## Characterization of ZnO-NPs

While the ZnO-NPs solution was being stirred, its UV-visible absorption spectra were measured using a UV-visible spectrophotometer (SPECORD 200 PLUS, An Endress+Hauser Company) in the scanning range of 300–600 nm,

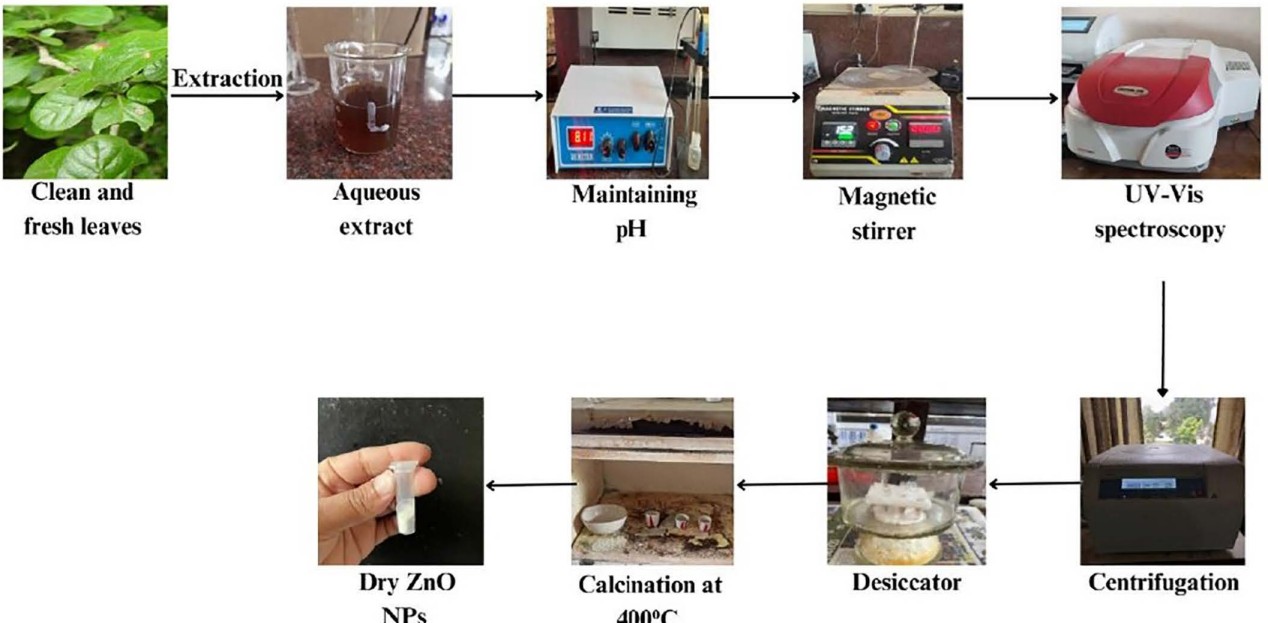

**Fig 3. Schematic route for the synthesis of ZnO-NPs.**

**Fig 4. The reaction mechanism shown for the use of metabolites in the synthesis of ZnO-NPs.**

with deionized water as the blank. Bio-synthesized ZnO-NPs were placed on a potassium bromide pellet for FTIR analysis and scanned with a Bruker Tensor 27 in the 400 $cm^{-1}$ to 4000 $cm^{-1}$ wavelength range to determine their functional group. A D2 phaser (Bruker, NAST, Nepal), a powerful X-ray beam diffraction analyzer, was used to study the produced nanoparticles further and determine their crystal structure. The size and shape of ZnO nanoparticles (ZnO-NPs) were studied using FE-SEM at 10 kV using SU-70 equipment in Korea. Energy-dispersive X-ray (EDX) analysis with EDAX APEX software was used to determine the various elemental compositions of the nanoparticles.

## Evaluation of antioxidant activity

The antioxidant potential ($IC_{50}$) is the amount necessary to block and evaluate its DPPH scavenging ability by half [28]. The DPPH radical scavenging experiment was utilized to assess antioxidant activity [29,30]. Nanoparticles (3.75 µg/mL from 100 µg/mL) and aqueous *C. Spinosa* leaf extract (100 µL) were combined with a methanolic solution of DPPH (100 µL, 0.1 mM) at various doses (500, 250, 125, 62.5, 31.25, and 15.625 µg/mL). After 30 minutes of dark incubation, the absorbance at 517 nm was measured. The reference standard was quercetin (20 µg/mL to 0.625 µg/mL), and each test was conducted in triplicate. Eq (1) was used to calculate the percentage of scavenging.

$$\text{Percentage scavenging } (\%) = \frac{(A control - A sample)}{A\ control} \times 100 \tag{1}$$

Where, A sample = absorbance of sample, A control = absorbance of control

## Evaluation of antimicrobial activity

The antimicrobial properties of ZnO-NPs were investigated in vitro using the Well diffusion method with reference antibiotic 1 mg/mL of neomycin following a specified procedure against three Gram-negative strains of *Shigella sonnei* (ATCC 25931), *Klebsiella pneumoniae* (ATCC 700603), and *Escherichia coli* (ATCC 25912), as well as one Gram-positive strain of *Staphylococcus aureus* (ATCC 43300) [31]. The testing pathogens were incubated at 37 °C after being injected into Muller-Hinton Broth (MHB). Upon bringing the turbidity down to the standard 0.5 McFarland, $1.5 \times 108$ CFU/mL was the final inoculum. A well holding 25 µL of the test sample (50 mg/mL) dissolved in 100% distilled water, a negative control (100% distilled water), and a positive control (1 mg/mL, neomycin) were all included in each experiment set. The contents were allowed to diffuse at room temperature for 15 minutes before being incubated at 37°C for 24 hours. The ZOI (mm) that surrounded the well was apparent on the plates following incubation. ZnO-NPs, controls, and crude extract were kept in separate wells in each plate for comparison. Thus, it is possible to compare and contrast whether the antibacterial activity is caused by nanoparticles, crude extracts, or controls. To create a strong enough inhibitory impact on the targeted bacteria to allow for clear observation of the antimicrobial agent's efficacy, high concentrations, such as 50 mg/mL of the tested sample, were used in antimicrobial experiments. This experiment is merely the first step in determining nanoparticles' antibacterial properties. MIC and MBC assays were used to further examine the lowest concentration of sample required for inhibition.

## Estimation of minimum inhibitory concentration (MIC) and minimum bactericidal concentration (MBC)

The Resazurin Microtiter Method (RMM) was used to measure MIC [32]. In a 96-well plate filled with Muller-Hinton broth (MHB), the test samples were serially diluted to achieve various concentrations. After calibrating to a turbidity of 0.5 McFarland, 5 µL of bacterial suspension was injected into each well (except the negative control). The well-known medication neomycin was used as the positive control since it is an aminoglycoside antibiotic that has been extensively tested and demonstrated to be effective against a wide range of bacteria. Because of its predictable and continuous antibacterial effect, it is an excellent tool for comparing the performance of various antimicrobial medications or determining the susceptibility of bacterial strains. After 24 hours of incubation at 37 °C, the microtiter plates were then incubated for 4 hours with 0.003% resazurin added. The wells with bacterial growth turned pink, whereas those with none remained blue. The contents of the wells were streaked onto nutrient agar plates to calculate the MBC, which was then incubated at 37 °C for 24 hours.

## Brine shrimp lethality assay

The toxicity was assessed using a modified brine shrimp lethality assay [33]. To create artificial saline water for the successful hatching of brine shrimp cysts, one liter of double-distilled water was mixed with 40.56 grams of salt. To maintain a pH between 8 and 8.5, 1N NaOH was utilized. Then, a 5 mL test solution comprising 4 mL of saline water and 1 mL of

the test sample at different concentrations (1000, 500, 250, 125, 100, and 10 μL) was administered to 10 freshly hatched nauplii for a full day. After one day, the number of live nauplii in each test tube was recorded. The mortality percentage (%) was determined using Eq (2).

$$M \ (\% \ \text{vs. control}) \ = \ \frac{(Nc - Nt)}{Nc} \times 100$$

(2)

Nc and Nt reflect the number of live nauplii in the control and tested agents, respectively, after 24 hours, while M denotes the mortality rate.

### Statistical analysis

GraphPad Prism 9.5.1 and MS Excel software were used to calculate the $IC_{50}$ in the DPPH and $LC_{50}$ in the brine shrimp lethality experiment, as well as to create the appropriate graphs. The tests were repeated three times, and the results were presented as mean ± standard deviation. Origin 2019b (64-bit) and High Score (Plus) were used to plot the FTIR, XRD, and UV-visible spectra. SEM pictures were analyzed using the ImageJ software.

## Results

### Structural and optical characteristics

When Zn $(NO_3)_2.6H_2O$ was added to the extract, it turned yellowish white, indicating that ZnO-NPs were formed. Following calcination, a white powder was generated, which is consistent with prior studies by [34]. Fig 5(a) depicts the effective synthesis of ZnO-NPs, with an absorbance peak for ZnO at 366 nm. Apart from indicating purity, the XRD peaks show the crystal structure of ZnO-NP, as shown in Fig 5(b). The peaks correspond to standard JCPDS card No. 01-079-0205 and are located at approximately 2θ values of 69.1°, 68.0°, 62.9°, 56.6°, 47.5°, 36.2°, 34.4°, and 31.7°. ZnO crystal planes (112), (200), (103), (110), (102), (101), (002), and (100) are shown by peaks with appropriate 2θ values. Strong, clear peaks with good crystallinity indicated that the ZnO-NPs had a well-organized crystalline structure with a crystallite size (D) of 12.39 ± 3.84 nm, as determined by Eq (3). The utilization of several peaks to calculate the average crystallite size, however, is most likely the reason for the significant standard deviation from the mean, as various peaks have varied full width at half maximum (FWHM) values and related Bragg angles.

$$D \ = \ \frac{k\lambda}{\beta cos\theta}$$

(3)

The dimensionless form factor ($\sim$1) is represented by k, the crystalline grain size by D, the full width at half maximum (FWHM) in radians by β, the Bragg's angle, or half of the 2θ value of the selected peak, by θ, and the wavelength of x-ray radiation by λ.

Numerous bands in the 4000–400 $cm^{-1}$ range revealed the samples' organic composition, as illustrated in Fig 5(c). Aqueous extract peaks were found at 3676, 2978, 2902, 2322, 2169, 2017, 1658, 1398, 1238, 1064, 879, and 671 $cm^{-1}$. FTIR peaks at 3676 $cm^{-1}$ are often formed by stretching and bending vibrations of hydroxyl groups (O-H), which are frequently related to the presence of alcohols, phenols, or carboxylic acids in plant extracts. These groups can produce ZnO nanoparticles by transforming zinc ions ($Zn^{2+}$) into zinc atoms ($Zn^0$) via electron donation [35]. The stretching vibration of aliphatic C-H groups is related to the peaks at 2978 $cm^{-1}$ and 2902 $cm^{-1}$. Peaks at 2322 $cm^{-1}$, 2017 $cm^{-1}$, 1392 $cm^{-1}$, and 1238 $cm^{-1}$ may suggest the presence of functional groups like C-O or C-N, which can act as capping or stabilizing agents during nanoparticle production. A peak at 2169 $cm^{-1}$ or 1658 $cm^{-1}$ represents an alkene C=C bond, amide I (C=O stretching), or esterified carboxyl groups. A peak at 879 $cm^{-1}$ is most likely associated with alkenes or alkynes, while a peak at

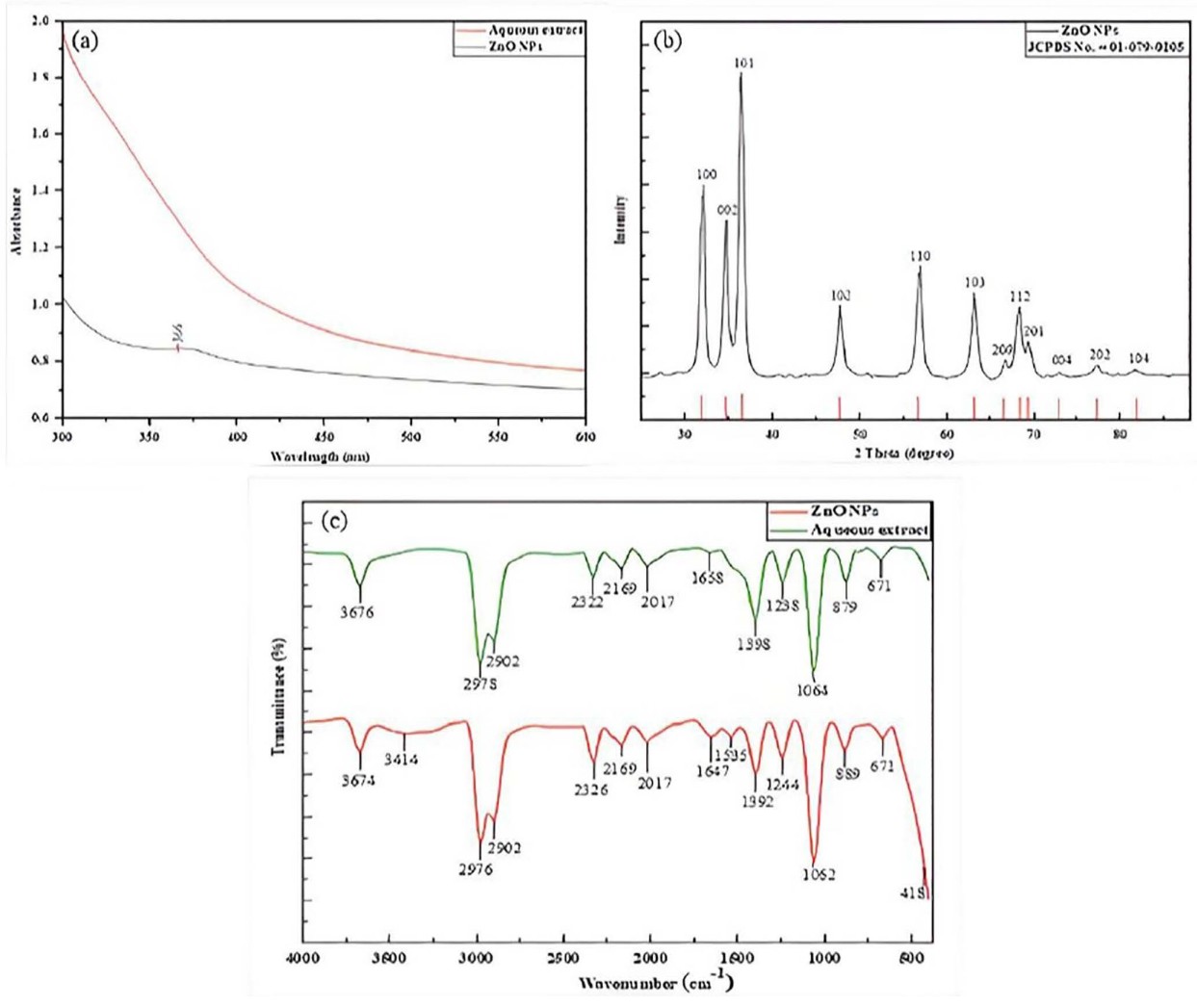

**Fig 5. Characterization of ZnO-NPs (a) UV-vis spectrum of aqueous extract and synthesized ZnO-NPs, (b) XRD pattern of synthesized ZnO-NPs, and (c) FTIR spectrum of aqueous extract and synthesized ZnO-NPs.**

$1064 \, cm^{-1}$ is associated with C-O stretching vibrations or aromatic ring vibrations. The FTIR spectra of ZnO-NPs showed similar peaks with varying intensities at 3674, 3414, 2976, 2902, 2326, 2169, 2017, 1647, 1535, 1392, 1244, 1062, 889, $671 \, cm^{-1}$, and $418 \, cm^{-1}$. The spectra revealed a clear Zn-O bonding peak around $671 \, cm^{-1}$ to $418 \, cm^{-1}$, similar to prior research [36].

## Morphological and compositional characteristics

Fig 6(a–d) depicts the results of an FE-SEM analysis that examined the surface form and elemental composition of the produced ZnO-NPs. In a previous study [25], spherical ZnO-NPs measuring 37.49 nm were produced utilizing an aqueous fruit extract of *Capparis spinosa* L. In this instance, FE-SEM revealed that the average size of the nearly spherical ZnO-NPs was 22.08 ± 0.36 nm, which is less than 100 nm. This disagreement could be caused by differences in several variables, such as plant species, location, sample collection time, extract to metal salt ratio, pH utilized, and other laboratory

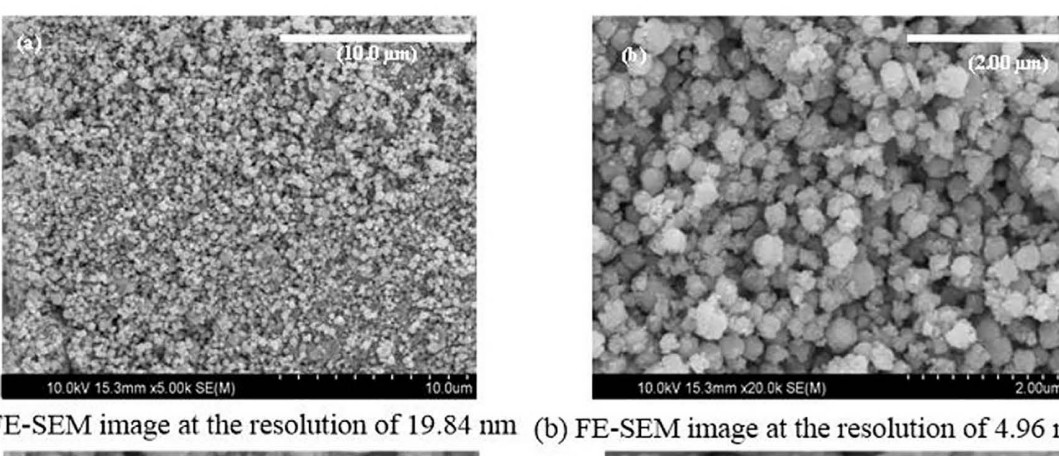

(a) FE-SEM image at the resolution of 19.84 nm (b) FE-SEM image at the resolution of 4.96 nm

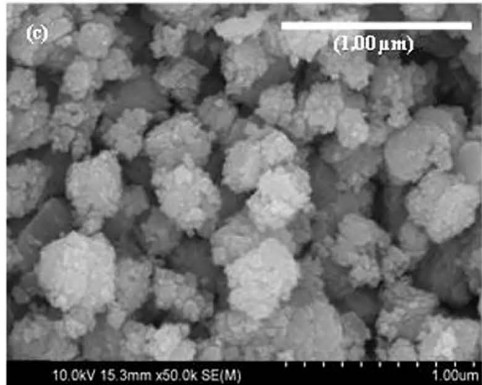
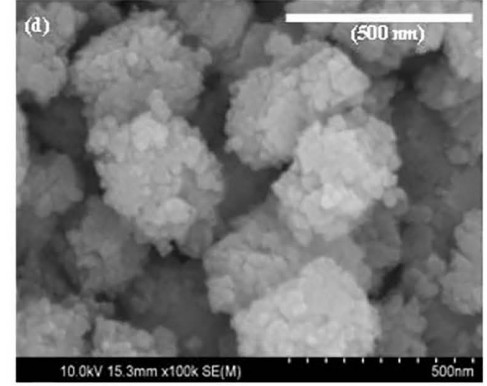

(c) FE-SEM image at the resolution of 0.99 nm (d) FE-SEM image at the resolution of 4.96 nm

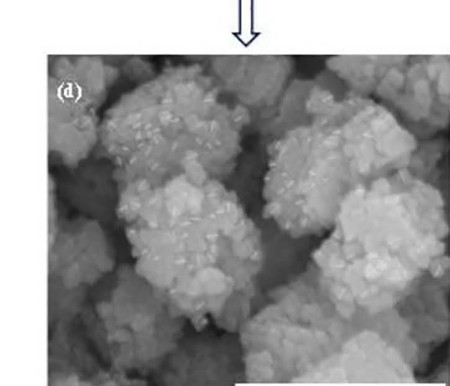

**Fig 6. FE-SEM images of synthesized ZnO-NPs by using aqueous leaf extract of *C. spinosa* at different resolutions (a) FE-SEM image at the resolution of 19.84 nm, (b) FE-SEM image at the resolution of 4.96 nm, (c) FE-SEM image at the resolution of 0.99 nm, and (d) FE-SEM image at the resolution of 4.96 nm.**

conditions, as stated by [37]. Fig 7 shows the elemental composition of ZnO-NPS, as well as the color mapping established by EDX testing. Following Fig 8(a), the EDX peaks provide strong evidence that ZnO-NPs were manufactured with 70.4% zinc and 18.3% oxygen. The sample also contained trace amounts of carbon and nitrogen. The histogram seen in Fig 8(b) illustrates the particle size distribution.

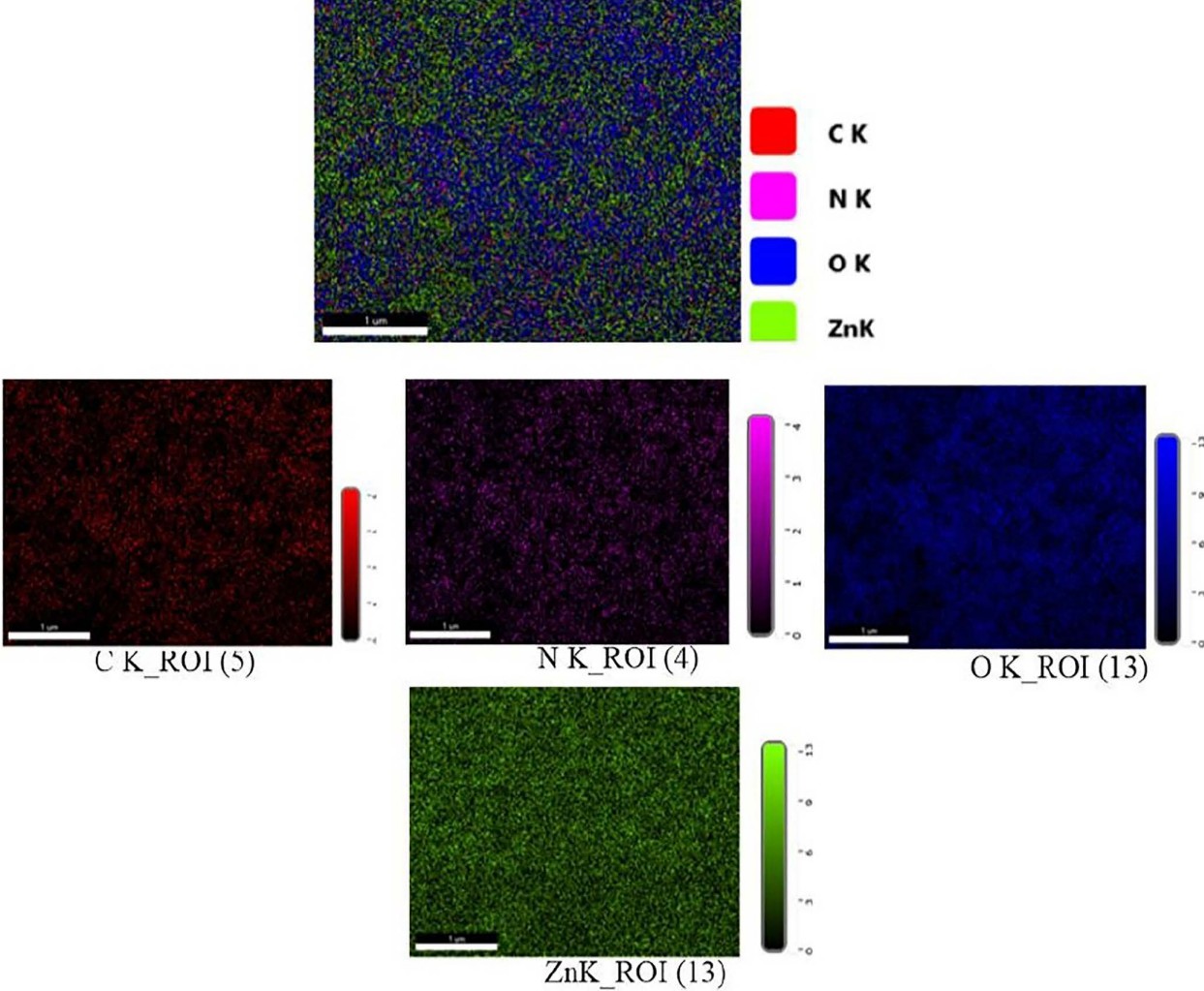

**Fig 7. EDX spectrum of Phytosynthesized ZnO-NPs by an aqueous leaf extract of *C. spinosa* with total elemental mapping and individual color distribution.**

## Antioxidant potential of nanoparticles

Fig 9 (a and b) graphically shows the inhibitory potential of the aqueous extract and standard quercetin. Fig 9 (c) illustrates the NPs' antioxidant capability. ZnO-NPs had a lower $IC_{50}$ (94.83 ± 0.00 µg/mL) than the plant extract, indicating improved antioxidant action over the aqueous extract. As the hue of the DPPH solution changed over time, the maximum intensity for the samples at 517 nm gradually dropped. The $IC_{50}$ values for produced ZnO-NPs, aqueous extracts, and standard quercetin are shown in Table 1.

## Antimicrobial activity

Table 2 and Fig 10 show that, in comparison to other bacteria such as *Shigella sonnei* (13 mm) and *Escherichia coli* (9 mm), ZnO-NPs inhibited *Klebsiella pneumoniae* (16 mm) and *Staphylococcus aureus* (15 mm) with the biggest zone of inhibition. *Escherichia coli* samples were the least susceptible of the four strains examined. As particle size decreased,

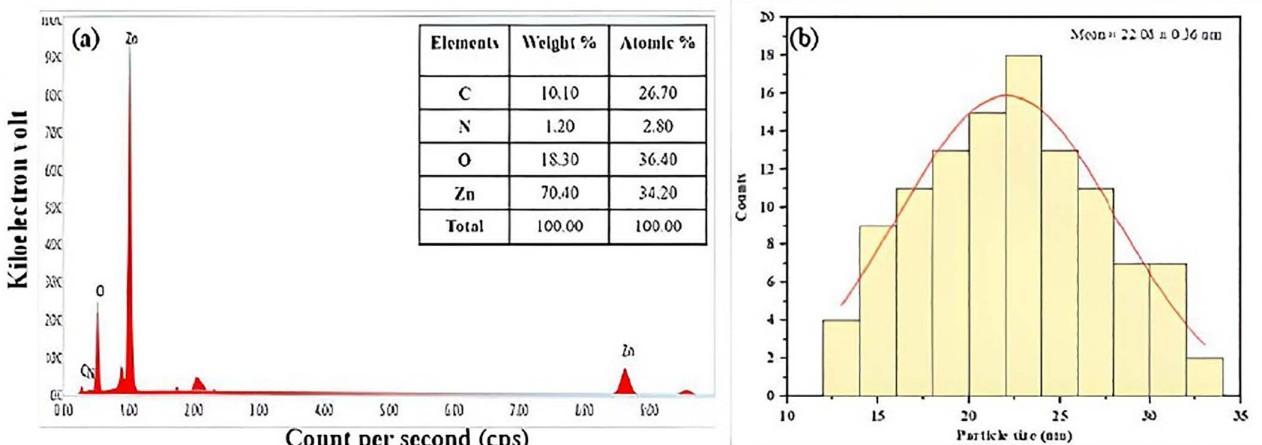

**Fig 8. Characterization of ZnO-NPs (a) EDX spectrum of ZnO-NPs and (b) FE-SEM images showing average particle size through a histogram for synthesized ZnO-NPs.**

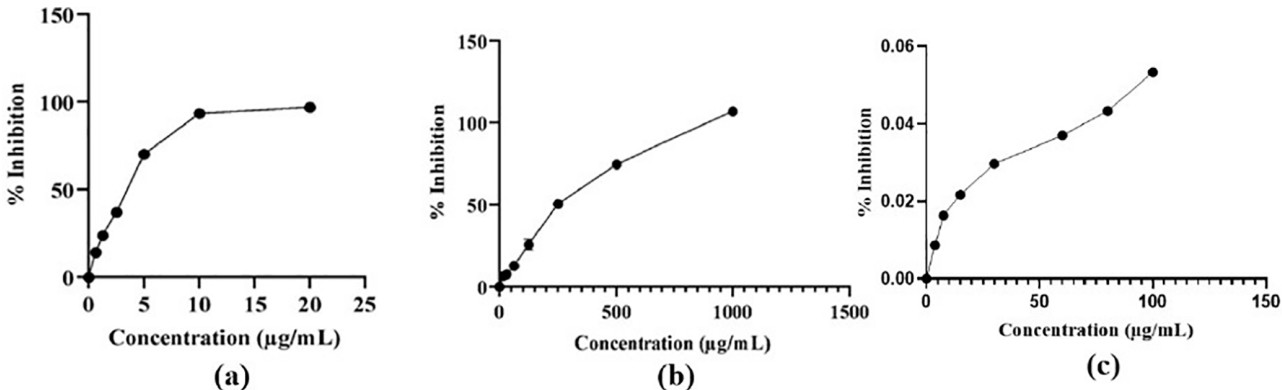

**Fig 9. (a) Standard curve of DPPH inhibition by quercetin, (b) plot of DPPH inhibition against different concentrations for aqueous extracts of the leaf of *C. spinosa*, and (c) A plot of DPPH inhibition against different concentrations of ZnO-NPs.**

**Table 1. IC$_{50}$ of standard quercetin, aqueous extracts, and ZnO-NPs.**

| S.N. | Sample used | IC$_{50}$ (µg/mL) |
|------|-------------|-------------------|
| 1 | quercetin | 3.83 ± 0.00 |
| 2 | LA | 723.20 ± 0.00 |
| 3 | ZnO-NPs | 94.83 ± 0.00 |

Where, ZnO-NPs = Zinc oxide nanoparticles, LA = leaf aqueous extract.

greater surface area permitted optimal NP dispersion in the media. This expedited the formation of surface oxygen species, eventually causing the pathogen to die by rupturing its membrane. The effectiveness of ZnO-NPs against bacteria may be explained by their direct interaction with pathogen cell walls, which disintegrates and releases antimicrobial ions ($Zn^{2+}$) [38]. As seen in Fig 11, all of the pathogens' vital bacterial functions are disrupted after these free ions connect

**Table 2. Antimicrobial activity shown by ZnO-NPs against four different pathogens.**

| Nanoparticles | Zone of inhibition (mm) | | | |
|---|---|---|---|---|
| | *Staphylococcus aureus* | *Shigella sonnei* | *Klebsiella pneumoniae* | *Escherichia coli* |
| LA | 10 | 10 | 10 | 9 |
| ZnO-NPs | 15 | 13 | 16 | 9 |
| Neomycin | 21 | 20 | 22 | 21 |

Where, ZnO-NPs = zinc oxide nanoparticles, LA = leaf aqueous extract.

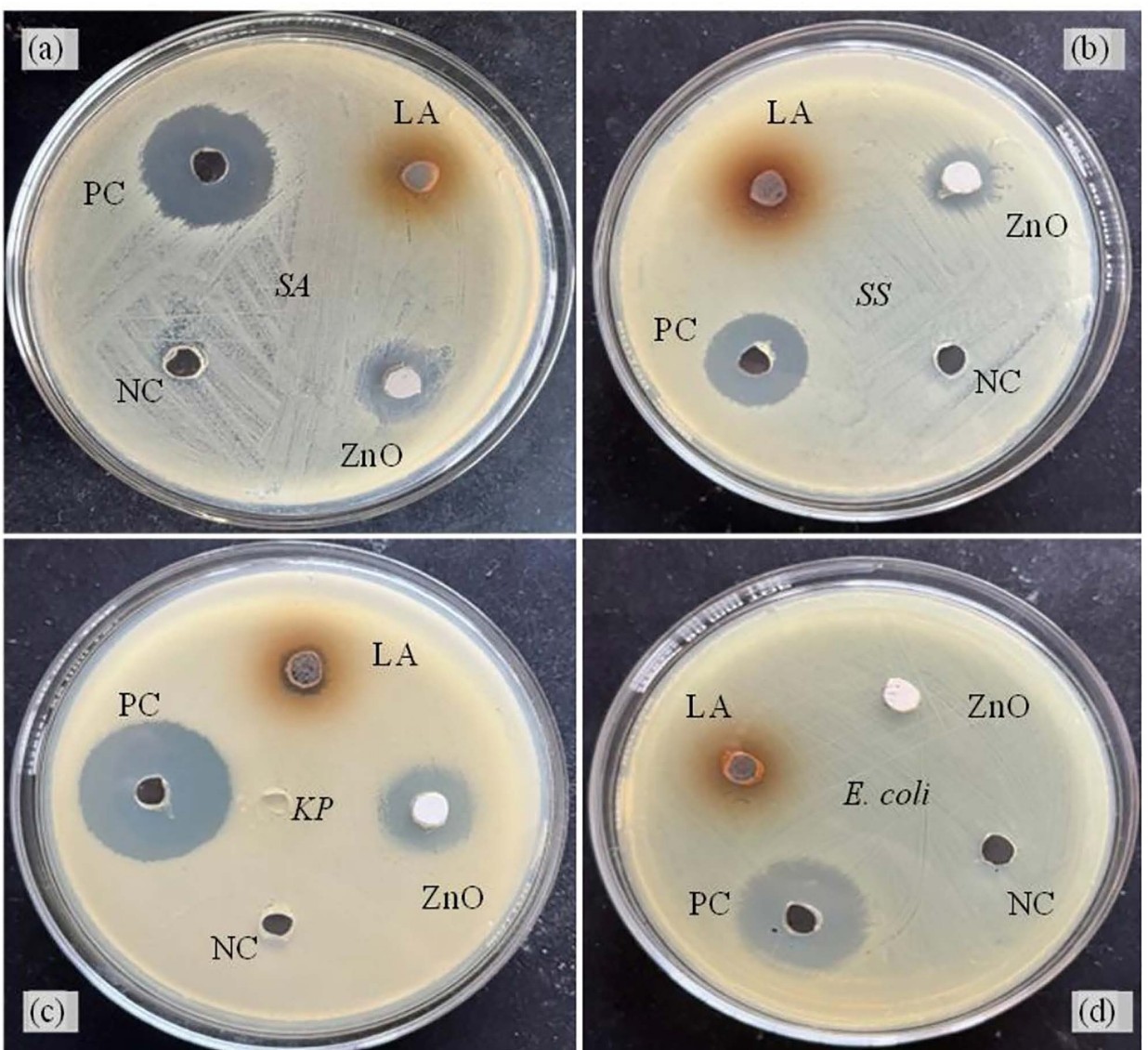

**Fig 10. ZOI shown by an aqueous extract and ZnO-NPs against four pathogenic microorganisms (a)** *Staphylococcus aureus*, **(b)** *Shigella sonnei*, **(c)** *Klebsiella pneumoniae*, **and (d)** *Escherichia coli*, **respectively.**

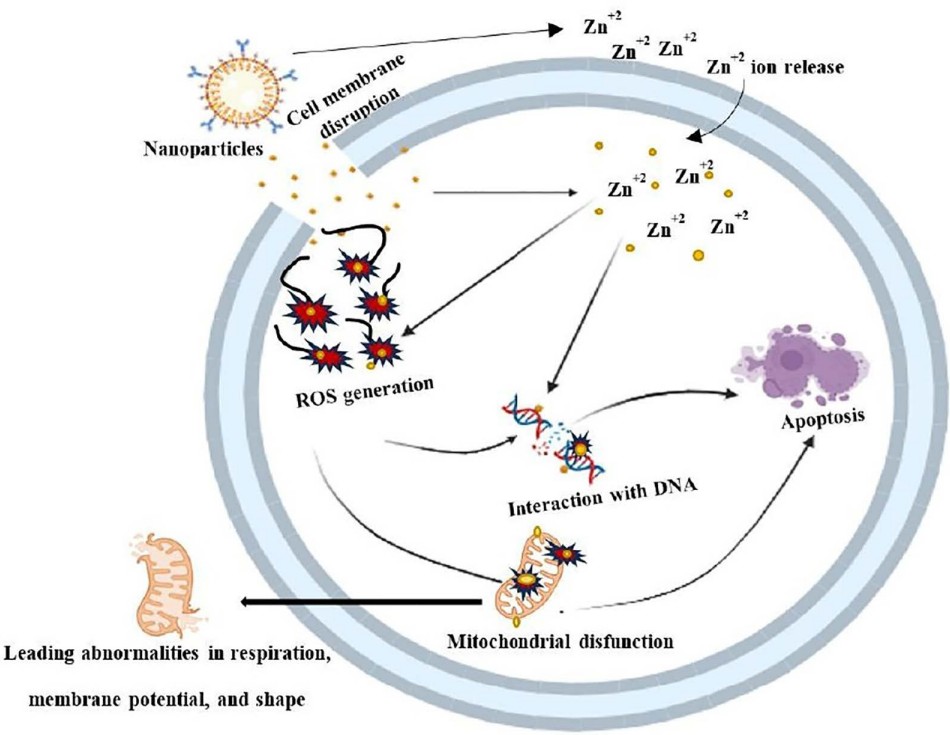

**Fig 11. Mechanism of antimicrobial activity shown by ZnO-NPs.**

to their proteins and polysaccharides. The results clearly show that the activity of the nanoparticles created is greater than that of the crude extract, whilst the negative control (water) has no effect on any of the microorganisms under investigation.

## Minimum inhibitory concentration (MIC) and minimum bactericidal concentration (MBC) of nanoparticles

After testing against four bacterial strains, the nanoparticles showed the most potent antibacterial action against one Gram-positive and one Gram-negative pathogen. This motivated further research into the MIC and MBC for these bacteria. NPs suppressed both bacterial strains at 50 µg/mL. Table 3 shows that for each strain chosen, ZnO-NPs had MBC and MIC of 12.5 mg/mL and 6.25 mg/mL, respectively. Fig 12(a–c) demonstrates the minimal inhibitory concentration and nutritional agar plates with the lowest bactericidal concentration of ZnO-NPs against *Staphylococcus aureus* and *Klebsiella pneumoniae*, respectively.

## Brine shrimp lethality assay

Over 24 hours, the average mortality rate of ZnO-NPs ranged from 40% (10 µg/mL) to 83.33% (1000 µg/mL). Similar to [39], the mortality rate rose with NP content. This indicates that ZnO-NPs were relatively toxic to *Artemia nauplii* throughout the course of a 24-hour exposure at the maximum test concentration. Table 4 shows the toxicity of ZnO-NPs produced from aqueous leaf extract, with an $LC_{50}$ of $55.20 \pm 16.19$ µg/mL. Fig 13 depicts the concentration and mortality percentage of *Artemia nauplii* caused by ZnO-NPs.

**Table 3. MIC and MBC shown by ZnO-NPs.**

| S.N. | Samples | MIC (mg/mL) | | MBC (mg/mL) | |
|---|---|---|---|---|---|
| | | Staphylococcus aureus | Klebsiella pneumoniae | Staphylococcus aureus | Klebsiella pneumoniae |
| 2. | ZnO-NPs | 6.25 | 6.25 | 12.50 | 12.50 |
| 3. | Neomycin (Positive control) | 0.031 | 0.004 | 0.063 | 0.008 |

Where, ZnO = nanoparticles, LA = leaf aqueous.

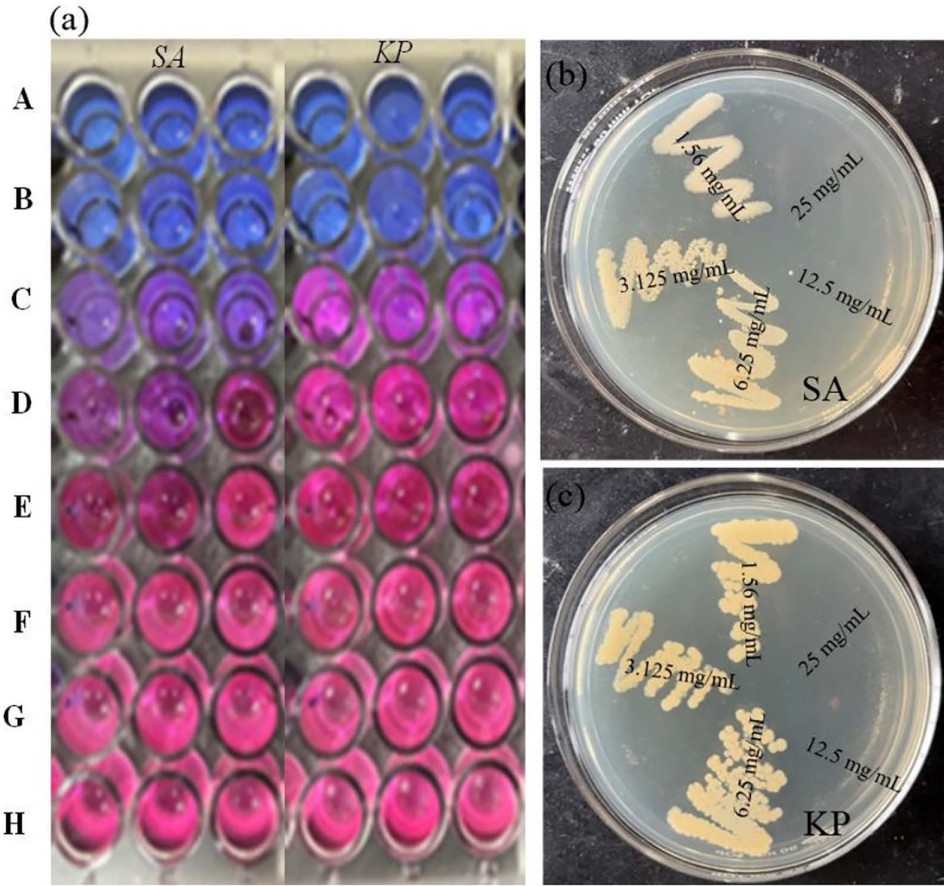

**Fig 12. (a)** Minimum inhibitory concentration shown by ZnO-NPs against *Staphylococcus aureus* and *Klebsiella pneumoniae* respectively, where the concentrations A=25mg/mL, B=12.5mg/mL, C=6.25mg/mL, D=3.125mg/mL, E=1.563mg/mL, F=0.781mg/mL, G=0.391mg/mL, and H=0.195mg/mL, and Nutrient agar plates showing Minimum bactericidal concentration shown by ZnO-NPs against **(b)** *Staphylococcus aureus,* and **(c)** *Klebsiella pneumoniae*.

## Discussion

The physicochemical factors of nanoparticles (NPs), such as size, chemical composition, and surface charge, are widely accepted to influence their properties. One of the key properties of nanoparticles is their small size. As more molecules are exposed, the surface area available for interaction with biomolecules increases [40,41]. The aqueous extract of *C.*

**Table 4. LC$_{50}$ values shown by ZnO-NPs.**

| S.N. | Nanoparticles | Regression equation | LC$_{50}$ (µg/mL) |
|------|---------------|---------------------|-------------------|
| 1. | ZnO NPs | y = 0.0412x + 46.384<br>R² = 0.8805 | 55.20 ± 16.19 |

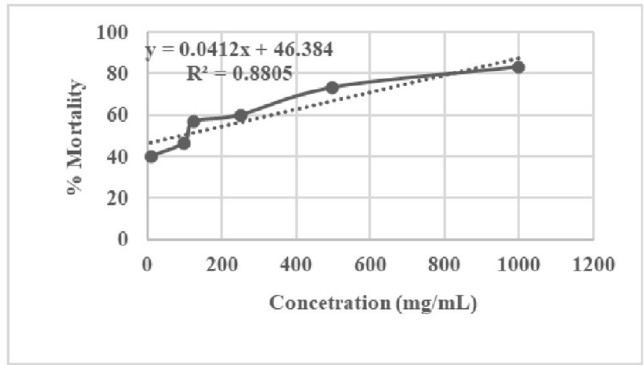

**Fig 13. Plot of concentration vs % mortality of *Artemia nauplii*.**

*spinosa* leaves was employed in this study to successfully produce zinc oxide nanoparticles (ZnO-NPs) in a cheap and environmentally friendly method.

Plants may have therapeutic qualities due to particular molecules or the synergistic interaction of numerous phytoconstituents that reduce Zn (NO$_3$)$_2$.6H$_2$O into ZnO during the formation of ZnO-NPs. The direct observation of a change in color in the solution containing plant extract and metal salt precursor provided initial proof of the formation of nanoparticles utilizing *C. spinosa* leaf extract.

The structural characterization methods, such as XRD, FE-SEM, EDX, FTIR, and UV-vis spectroscopy, verified the development of ZnO-NPs. The nanoparticles have a 3.24 eV band gap energy and a clear absorption peak at 366 nm. Green-synthesized ZnO-NPs exhibited similar peaks at 334 and 374 nm [25,42]. The crystallite size of nanoparticles was determined using the Debye-Scherrer formula, and the result was 12.39 ± 3.84 nm, comparable to *Artocarpus hirsutus* [43] and *El Salvador persica* [44].

The interaction between ZnO-NPs and functional groups was found to be responsible for the small shift and change in the position and intensity of multiple connected peaks in the NPs' FTIR spectra [45]. The FTIR spectrum of ZnO-NPs also includes a peak at a wavelength of 418 cm$^{-1}$, suggesting vibrational strain absorption of the Zn-O connection and confirming the effective production of ZnO particles. The functional groups in *C. spinosa* leaf extract, including (O-H), (C≡N), and (C=O), can stabilize, reduce, and cap NPs during biosynthesis. Capping agents appear to use a variety of processes to stabilize NPs, including van der Waals forces, hydration force stabilization, steric stabilization, and electrostatic stability [46]. The results are similar to ZnO-NPs prepared with extracts from *Acacia catechu*, *Artemisia vulgaris*, and *Cynodon dactylon* previously described by [47].

The SEM image analysis revealed that the NPs were agglomerated and comparably spherical. A SEM histogram revealed an average particle size of 22.08 ± 0.36 nm. The EDX data validated the production of ZnO-NPs. Other elements, such as carbon and nitrogen, were present in the produced particles. Similar to the current study, [48] verified that the NPs were spherical with sizes of 25.14 ± 7.4 nm via SEM analysis.

ZnO-NPs had a high antioxidant activity, with an $IC_{50}$ of 94.83±0.00 µg/mL, as measured by a DPPH experiment. The antioxidant activity of ZnO-NPs rose as concentration increased; however, it remained below the recommended level, i.e., standard as in the research published [49,50]. Using the DPPH test, [50] also evaluated the ZnO-NPs' capacity to scavenge radicals caused by *Moringa oleifera* extract, in which ZnO-NPs concentration was directly related to radical scavenging activity, peaking at 67% at 100 µg/mL.

The generated ZnO-NPs were efficient against *Staphylococcus aureus* and *Klebsiella pneumoniae* at ZOIs of 15 mm and 16 mm, respectively. The MIC and MBC values of ZnO-NPs against *Staphylococcus aureus* and *Klebsiella pneumoniae* were 6.25 mg/mL and 12.5 mg/mL, respectively. This suggests that the nanoparticles inhibit Gram-positive and Gram-negative bacteria at roughly the same rate. ZnO-NPs were found to have antibacterial activity against *Aspergillus terreus* (18.00±0.50), *Aspergillus niger* (19.00±1.00), and *Staphylococcus aureus* (11.00±0.50) in a similar study [51]. In addition, [52] tested 28 isolates for MIC and MBC. The MIC and MBC of ZnO-NPs were 31.25 µg/mL and 62.5 µg/mL, respectively.

*Artemia salina* demonstrated enhanced catalase (CAT) activity in a study by [53], implying that NPs' interaction may alter the equilibrium of aquatic species. They can withstand ambient levels of toxins, notably metal oxides, while less resistant species, such as zebrafish, may suffer if the concentration exceeds *Artemia salina*'s $LC_{50}$ [54]. In the toxicity assay, NPs were found to be more effective, with an $LC_{50}$ value of 55.20±16.19 µg/mL. As said by [55], nanoparticles with an $LC_{50}$ of less than 100 µg/mL are more potent. Furthermore, at different concentrations, ZnO-NPs demonstrated variable mortality rates against *Artemia salina*, which is compatible with the findings of [56]. Similarly, previous studies on ZnO-NPs biosynthesized from *Ficus racemosa* L. leaf extract showed reduced acute toxicity to brine shrimp after 48 hours of treatment [57]. In summary, these data reveal that ZnO-NP derived from *C. spinosa* aqueous leaf extract has excellent antibacterial, antioxidant, and toxic properties that could be tremendously beneficial in biomedicine. Because the chemistry of NPs is complex, further research is needed to properly understand their potential effects on human health and the environment.

## Conclusions

This study describes the green synthesis of ZnO-NPs with *C. spinosa* leaf extract as a reducing agent. UV-vis, FTIR, and EDX characterizations were carried out to validate the formation of ZnO-NPs. XRD and FE-SEM were used to calculate the average particle size of the nanoparticles. Antioxidant assays revealed strong free radical scavenging activity at higher concentrations, whereas antibacterial studies revealed significant inhibitory effects, particularly against *Klebsiella pneumoniae* and *Staphylococcus aureus*, indicating that the nanoparticles inhibit both Gram-positive and Gram-negative bacteria. ZnO-NPs exhibited dose-dependent toxicity against *Artemia salina*, becoming more toxic at higher doses (1000 µg/mL) and less dangerous at lower levels (10 µg/mL). Certain findings may be comparable to the literature since the plant sample contains similar compounds, but the results vary from previous research for a variety of reasons, including plant species, geography, collection period, sample size, and other factors. All things considered, this study successfully connects plant bioactive chemicals to the creation and functionality of nanoparticles, thereby providing a sustainable and biologically relevant platform for future nanobiotechnology applications.

## Future prospects

These findings demonstrate that biogenic ZnO-NPs derived from *C. Spinosa* have a wide range of potential biomedical uses, promoting green chemistry and sustainable nanotechnology. Thus, future studies should concentrate on increasing synthesis volume, improving reaction conditions, measuring cytotoxicity and in vivo effectiveness, and investigating more untapped therapeutic potentials for producing ecologically friendly nanoparticles.

## Supporting information

**S1 Fig. Graphical abstract.**

(JPG)

## Acknowledgments

We would like to express our sincere gratitude to the National Herbarium and Plant Laboratories, Godawari, Lalitpur, Nepal, for the identification of the plant. We are grateful to the Nepal Academy of Science and Technology (NAST), Lalitpur, Nepal, for the XRD data. The Institute of Biomolecule Reconstruction at Sun Moon University, Republic of Korea, for supplying the bacterial strains.

## Author contributions

**Conceptualization:** Khaga Raj Sharma.

**Data curation:** Khaga Raj Sharma, Rabina Baraili, Ishwor Pathak, Sugam Sharma, Manisha Bhusal.

**Formal analysis:** Rabina Baraili, Sugam Sharma, Manisha Bhusal.

**Software:** Ishwor Pathak, Sugam Sharma.

**Supervision:** Khaga Raj Sharma.

**Writing – original draft:** Rabina Baraili.

**Writing – review & editing:** Khaga Raj Sharma.

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
