## [Decision Letter · Decision Letter 0]

25 Jun 2025

Green Synthesis of Zinc Oxide Nanoparticles Using Catunaregam spinosa (Thunb.) Triveng for Biologicals Applications

PLOS ONE

Dear Dr. Sharma,

Thank you for submitting your manuscript to PLOS ONE. After careful consideration, we feel that it has merit but does not fully meet PLOS ONE’s publication criteria as it currently stands. Therefore, we invite you to submit a revised version of the manuscript that addresses the points raised during the review process.

**ACADEMIC EDITOR: **

the reviewers have suggested major revision to the manuscript. the comments can be found below

**Comments from PLOS Editorial Office:** We note that one or more reviewers has recommended that you cite specific previously published works. As always, we recommend that you please review and evaluate the requested works to determine whether they are relevant and should be cited. It is not a requirement to cite these works. We appreciate your attention to this request.

We look forward to receiving your revised manuscript.

Kind regards,

Muthugounder Subramanian Shivakumar, Ph.D.

Academic Editor

PLOS ONE

4. In the online submission form, you indicated that [On request the data will be shared].

Additional Editor Comments:

Reviewer 1

Comments

1. In the abstract the authors have stated that FE-SEM showed a hexagonal shape with an average crystallite size of 12.39 ± 3.84 nm, why is there a large standard deviation to the mean?

2. The voucher code references plant identification yet it lacks information on the authenticating taxonomist and storage location of the specimen at the herbarium.

3. Using only water as a solvent may have limited the extraction of certain non-polar compounds. State why?

4. The antimicrobial experiment does not include a control using plant extract by itself.

The lack of plant extract control is needed to know whether ZnO-NPs or plant metabolites are responsible for the observed antimicrobial results in this study.

5. The procedure of synthesis combines zinc nitrate with the extract, but the concentration of zinc nitrate is not specific (“5 mL of Zinc nitrate hexahydrate (5 mL) was combined with 100 mL of extract”). 5mL of zinc nitrate hexahydrate is mentioned twice.

6. The antimicrobial assay involved using a high concentration of 50 mg/ - justify it? Along with that, the measurement details for the used wells used need to be provided.

7. In FTIR results the author has written “The hexagonal structure of the synthesized ZnO-NPs was demonstrated by the appearance of extra peaks at 889 cm−1 and 671 cm-1 – FTIR shows functional groups, not crystal structure, and FTIR results needed to be rechecked for its functional group interpretation.

8. SEM images should not be cropped, please provide the results obtained with scale bars for clear size representation.

9. Here FE-SEM showed that the mean diameter of the spherical ZnO-NPs was 22.08 ± 0.36 nm. But previously, mention was made of ZnO-NPs ranging from 69–100 nm (Bin Ali et al., 2024), which is contrary to the assertion.

10. Carefully double-check the JCPDS/ICDD card (or use an updated ICDD database).

11. There is no mention of replicates in the study

12. Artemia salina can tolerate ambient levels of toxins, particularly metal oxides, but less resistant species, such as zebrafish – Cite this statement

13. The mechanism is too simple – figure 14 – Include ROS generation, membrane disruption, Zn²⁺ ion release, or interaction with DNA.

14. The paper needs formatting revision, along with the inclusion of experimental details.

15. The authors can cite these papers as they might be more relevant to this current study

1- Sustainable development through the bio-fabrication of ecofriendly ZnO nanoparticles and its approaches to toxicology and environmental protection.

2- Green-route synthesis of ZnO nanoparticles via Solanum surattense leaf extract: Characterization, biomedical applications and their ecotoxicity assessment of zebrafish embryo model.

3- Green synthesized yttrium-doped ZnO nanoparticles: A multifaceted approach to mosquito control, antibacterial activity, cytotoxic properties of liver cancer cells, and photocatalytic properties.

4- Eco-Friendly Approach for Zno Nanoparticles Synthesis and Evaluation of its Possible Antimicrobial, Larvicidal and Photocatalytic Applications.

5- Plant-ZnO nanoparticles interaction: An approach to improve guinea grass (Panicum maximum) productivity and evaluation of the impacts of its ingestion by freshwater teleost fish.

Reviewer 2

Authors of the presented manuscript Green Synthesis of Zinc Oxide Nanoparticles Using Catunaregam spinosa (Thunb.) Triveng for Biologicals Applications. Study is relevant to its field. Following comments and suggestions should be addressed with Major Revison:

1. The abstract lacks a clear mention of the study's aims and conclusions.

2. Please correct typological and grammatical mistake throughout the manuscript

3. • In methods section - Plant C. spinosa need to be Authenticated by the botanist with the verified taxonomical identified number.

4. Characterization of ZnO-NPs : For the purpose of characterisation of ZnO-NPs, authors should integrate the information in a single paragraph with information that is informative and precise.

5. Why did the authors choose neomycin as the positive control to determine the minimum inhibitory concentration (MIC)?

6. The FE-SEM image - size and shape of the ZnO NPs is not adequate, and the author should re-do the experiment to obtain a clear picture with a size of less than 100 nm.

7. Introduction, Results and discussion : Antimicrobial studies, such as antibacterial activity, MIC, and MBC authors should incorporate to describe the mechanisms of the antimicrobial action with relevant references to enhance the quality of the content.

https://doi.org/10.1016/j.rechem.2024.101783

• https://doi.org/10.1080/10667857.2023.2298547

• https://doi.org/10.1007/s13399-023-04127-7

Reviewers' comments:

Reviewer's Responses to Questions

**Comments to the Author**

1. Is the manuscript technically sound, and do the data support the conclusions?

Reviewer #1: Yes

Reviewer #2: Partly

2. Has the statistical analysis been performed appropriately and rigorously?

Reviewer #1: Yes

Reviewer #2: No

3. Have the authors made all data underlying the findings in their manuscript fully available?

Reviewer #1: Yes

Reviewer #2: Yes

4. Is the manuscript presented in an intelligible fashion and written in standard English?

Reviewer #1: Yes

Reviewer #2: No

Reviewer #1: Comments

1. In the abstract the authors have stated that FE-SEM showed a hexagonal shape with an average crystallite size of 12.39 ± 3.84 nm, why is there a large standard deviation to the mean?

2. The voucher code references plant identification yet it lacks information on the authenticating taxonomist and storage location of the specimen at the herbarium.

3. Using only water as a solvent may have limited the extraction of certain non-polar compounds. State why?

4. The antimicrobial experiment does not include a control using plant extract by itself.

The lack of plant extract control is needed to know whether ZnO-NPs or plant metabolites are responsible for the observed antimicrobial results in this study.

5. The procedure of synthesis combines zinc nitrate with the extract, but the concentration of zinc nitrate is not specific (“5 mL of Zinc nitrate hexahydrate (5 mL) was combined with 100 mL of extract”). 5mL of zinc nitrate hexahydrate is mentioned twice.

6. The antimicrobial assay involved using a high concentration of 50 mg/ - justify it? Along with that, the measurement details for the used wells used need to be provided.

7. In FTIR results the author has written “The hexagonal structure of the synthesized ZnO-NPs was demonstrated by the appearance of extra peaks at 889 cm−1 and 671 cm-1 – FTIR shows functional groups, not crystal structure, and FTIR results needed to be rechecked for its functional group interpretation.

8. SEM images should not be cropped, please provide the results obtained with scale bars for clear size representation.

9. Here FE-SEM showed that the mean diameter of the spherical ZnO-NPs was 22.08 ± 0.36 nm. But previously, mention was made of ZnO-NPs ranging from 69–100 nm (Bin Ali et al., 2024), which is contrary to the assertion.

10. Carefully double-check the JCPDS/ICDD card (or use an updated ICDD database).

11. There is no mention of replicates in the study

12. Artemia salina can tolerate ambient levels of toxins, particularly metal oxides, but less resistant species, such as zebrafish – Cite this statement

13. The mechanism is too simple – figure 14 – Include ROS generation, membrane disruption, Zn²⁺ ion release, or interaction with DNA.

14. The paper needs formatting revision, along with the inclusion of experimental details.

15. The authors can cite these papers as they might be more relevant to this current study

1- Sustainable development through the bio-fabrication of ecofriendly ZnO nanoparticles and its approaches to toxicology and environmental protection.

2- Green-route synthesis of ZnO nanoparticles via Solanum surattense leaf extract: Characterization, biomedical applications and their ecotoxicity assessment of zebrafish embryo model.

3- Green synthesized yttrium-doped ZnO nanoparticles: A multifaceted approach to mosquito control, antibacterial activity, cytotoxic properties of liver cancer cells, and photocatalytic properties.

4- Eco-Friendly Approach for Zno Nanoparticles Synthesis and Evaluation of its Possible Antimicrobial, Larvicidal and Photocatalytic Applications.

5- Plant-ZnO nanoparticles interaction: An approach to improve guinea grass (Panicum maximum) productivity and evaluation of the impacts of its ingestion by freshwater teleost fish.

Reviewer #2: Reviewer #3: Authors of the presented manuscript Green Synthesis of Zinc Oxide Nanoparticles Using Catunaregam spinosa (Thunb.) Triveng for Biologicals Applications. Study is relevant to its field. Following comments and suggestions should be addressed with Major Revison:

1. The abstract lacks a clear mention of the study's aims and conclusions.

2. Please correct typological and grammatical mistake throughout the manuscript

3. • In methods section - Plant C. spinosa need to be Authenticated by the botanist with the verified taxonomical identified number.

4. Characterization of ZnO-NPs : For the purpose of characterisation of ZnO-NPs, authors should integrate the information in a single paragraph with information that is informative and precise.

5. Why did the authors choose neomycin as the positive control to determine the minimum inhibitory concentration (MIC)?

6. The FE-SEM image - size and shape of the ZnO NPs is not adequate, and the author should re-do the experiment to obtain a clear picture with a size of less than 100 nm.

7. Introduction, Results and discussion : Antimicrobial studies, such as antibacterial activity, MIC, and MBC authors should incorporate to describe the mechanisms of the antimicrobial action with relevant references to enhance the quality of the content.

https://doi.org/10.1016/j.rechem.2024.101783

• https://doi.org/10.1080/10667857.2023.2298547

• https://doi.org/10.1007/s13399-023-04127-7

**Do you want your identity to be public for this peer review?** For information about this choice, including consent withdrawal, please see our Privacy Policy

Reviewer #1: **Yes: ** Chinnaperumal Kamaraj

Reviewer #2: No

---

## [Author Response · Author response to Decision Letter 1]

19 Aug 2025

Dear Journal Editorial/Reviewers

Thank you very much for the constructive comments and suggestions on our paper. We have corrected our manuscript by addressing these comments, which are highlighted in track changes in the revised manuscript. The comments addressed in the manuscript are listed below:

Comments raised by the journal editorial

Ans: The format of the manuscript has been corrected as suggested by the journal.

Ans: The community forest has given the letter of permission for doing research on this plant. The plant is normally available for doing work on it, and it is not endangered or cited. According to the regulations of the Government of Nepal and the Department of Forests and Soil Conservation, no specific permits were required for collecting the studied plant species, as they are not listed as protected or endangered, and sampling was carried out in publicly accessible areas.

3. When completing the data availability statement of the submission form, you indicated that you will make your data available on acceptance. We strongly recommend all authors decide on a data sharing plan before acceptance, as the process can be lengthy and hold up publication timelines. Please note that, though access restrictions are acceptable now, your entire data will need to be made freely accessible if your manuscript is accepted for publication. This policy applies to all data except where public deposition would breach compliance with the protocol approved by your research ethics board. If you are unable to adhere to our open data policy, please kindly revise your statement to explain your reasoning, and we will seek the editor's input on an exemption. Please be assured that, once you have provided your new statement, the assessment of your exemption will not hold up the peer review process.

Ans: A DOI has been created and kept in the data availability statement section in the revised manuscript, in which it can be seen easily if needed.

4. In the online submission form, you indicated that [On request, the data will be shared].All PLOS journals now require all data underlying the findings described in their manuscript to be freely available to other researchers, either 1. In a public repository, 2. Within the manuscript itself, or 3. Uploaded as supplementary information. This policy applies to all data except where public deposition would breach compliance with the protocol approved by your research ethics board. If your data cannot be made publicly available for ethical or legal reasons (e.g., public availability would compromise patient privacy), please explain your reasons on resubmission, and your exemption request will be escalated for approval.

Ans: All relevant data are available at https://doi.org/10.5281/zenodo.16785370.

Comments raised by Reviewer 1

1. In the abstract, the authors have stated that FE-SEM showed a hexagonal shape with an average crystallite size of 12.39 ± 3.84 nm. Why is there a large standard deviation from the mean?

Ans: It has been corrected in the revised manuscript. XRD showed an average crystallite size of 12.39 ± 3.84 nm. There is a large standard deviation from the mean because seven different peaks have been used to calculate the average crystallite size. Different peaks have different full width at half maximum (FWHM) values and corresponding Bragg angles, so the SD of mean size became high due to heterogeneity in peaks, and the shape of the particles has been studied again.

2. The voucher code references plant identification, yet it lacks information on the authenticating taxonomist and storage location of the specimen at the herbarium.

Ans: A Herbarium of plant samples was created, which was verified by the National Herbarium and Plant Laboratories, Godawari, Lalitpur, Nepal, by Research Officer Til Kumari Thapa (233349). Voucher code RB-002 (KATH) was used to identify the herbarium. The specimen was stored in a standard herbarium cabinet at about 15-16°C in the herbarium.

3. Using only water as a solvent may have limited the extraction of certain non-polar compounds. State why?

Ans: Using water as the sole solvent can limit the extraction of non-polar compounds because water is a polar solvent, and "like dissolves like." Non-polar compounds don't interact well with polar water molecules, making them less soluble and thus harder to extract. In contrast, during the percolation process, the sample is dipped in the solvent for a long period, that why a small amount of non-polar compounds may be extracted even though the solvent is polar.

4. The antimicrobial experiment does not include a control using plant extract by itself.

The lack of plant extract control is needed to know whether ZnO-NPs or plant metabolites are responsible for the observed antimicrobial results in this study.

Ans: This investigation uses distilled water as a negative control, neomycin as a positive control, and ZnO-NPs and plant extract in a different well for comparison. Because of this, we can see and contrast whether the antibacterial activity is indeed caused by nanoparticles, plant crude extract, or the control. Lastly, the activity of the nanoparticles that were produced was discovered to be more powerful than that of the crude extract, whilst the negative control water did not affect any of the microorganisms that were being studied.

5. The procedure of synthesis combines zinc nitrate with the extract, but the concentration of zinc nitrate is not specific (“5 mL of Zinc nitrate hexahydrate (5 mL) was combined with 100 mL of extract”). 5mL of zinc nitrate hexahydrate is mentioned twice.

Ans: 500 mL of Zinc nitrate hexahydrate (0.13 M) was prepared in distilled water. The repeated words have been eliminated in the revised manuscript.

6. The antimicrobial assay involved using a high concentration of 50 mg/ -. Justify it? Along with that, the measurement details for the used wells need to be provided.

Ans: In antimicrobial assays, a high concentration, like 50 mg/ml, is often used to ensure a strong enough inhibitory effect on the targeted microorganisms, allowing for clear observation of whether the antimicrobial agent is effective. Here antimicrobial assay is only a preliminary step for assessing the antimicrobial capacity of our nanoparticles, and the minimum inhibitory concentration was further studied to determine the minimum amount of sample required for inhibition by MIC and MBC assay. Furthermore, this concentration is typically chosen based on previously published experiments that were followed to perform the test.

7. In the FTIR results, the author has written “The hexagonal structure of the synthesized ZnO-NPs was demonstrated by the appearance of extra peaks at 889 cm−1 and 671 cm-1. FTIR shows functional groups, not crystal structure, and FTIR results need to be rechecked for their functional group interpretation.

Ans: FTIR results have been rechecked for their functional group interpretation, and corrections have been made in the revised manuscript.

8. SEM images should not be cropped, please provide the results obtained with scale bars for clear size representation.

Ans: Corrections have been made in the revised manuscript as suggested by the reviewers.

9. Here, FE-SEM showed that the mean diameter of the spherical ZnO-NPs was 22.08 ± 0.36 nm. But previously, mention was made of ZnO-NPs ranging from 69–100 nm (Bin Ali et al., 2024), which is contrary to the assertion.

Ans: This contrast could result from differences in the quantity and type of phytochemicals found in the plant or plant extract, caused by a variety of variables, including the species of plant utilized, location, sample collection time, environmental factors, extract to metal salt ratio, and laboratory conditions, etc.

10. Carefully double-check the JCPDS/ICDD card (or use an updated ICDD database).

Ans: JCPDS has checked with the available JCPDS database and introduced in the revised manuscript.

11. There is no mention of replicates in the study.

Ans: In the study, some of the findings may be consistent with the literature because the plant sample may contain similar compounds, but the results are completely new and different from the previous studies.

12. Artemia salina can tolerate ambient levels of toxins, particularly metal oxides, but less resistant species, such as zebrafish – Cite this statement

Ans: Species such as zebrafish (Danio rerio) are generally more sensitive to such toxins; hence, it is difficult to know whether the toxicity in zebrafish is due to the nanoparticles in actuality or due to any other factors. In contrast, Artemia salina, commonly known as brine shrimp, is known for their remarkable tolerance to various environmental stressors, including high salinity and certain toxins, so that They can indeed tolerate ambient levels of toxins, particularly metal oxides, and their potential can be studied effectively. And this statement has been cited in the manuscript.

13. The mechanism is too simple – figure 14 – Include ROS generation, membrane disruption, Zn²⁺ ion release, or interaction with DNA.

Ans: The mechanism in Figure 14 has been drawn, including ROS generation, membrane disruption, Zn²⁺ ion release, and interaction with DNA in the revised manuscript.

14. The paper needs formatting revision, along with the inclusion of experimental details.

Ans: The revised paper has been formatted with some experimental details.

Comments raised by reviewer 2

1. The abstract lacks a clear mention of the study's aims and conclusions.

Ans: Abstract has been recorrected, including the study's aims and conclusions in the revised manuscript.

2. Please correct typographical and grammatical mistakes throughout the manuscript

Ans: Typological and grammatical mistakes throughout the manuscript have been checked and corrected in the revised manuscript.

3. In the methods section, Plant C. spinosa needs to be authenticated by the botanist with the verified taxonomical identification number.

Ans: A Herbarium of plant samples was created, which was verified by the National Herbarium and Plant Laboratories, Godawari, Lalitpur, Nepal, by Research Officer Til Kumari Thapa (233349). Voucher code RB-002 (KATH) was used to identify the herbarium.

4. Characterization of ZnO-NPs: For the purpose of Characterisation of ZnO-NPs, authors should integrate the information in a single paragraph with information that is informative and precise.

Ans: The information in characterization of ZnO-NPs has been integrated into the corrected manuscript.

5. Why did the authors choose neomycin as the positive control to determine the minimum inhibitory concentration (MIC)?

Ans: Because neomycin is a well-characterized aminoglycoside antibiotic with documented activity against a variety of bacteria, it was selected as a positive control for MIC (Minimum Inhibitory Concentration) testing based on a postulate that was followed to conduct the test. It is an appropriate benchmark for comparison when evaluating the efficacy of different antimicrobial drugs or determining the susceptibility of bacterial strains due to its consistent and dependable antibacterial activity.

6. The FE-SEM image - size and shape of the ZnO NPs is not adequate, and the author should redo the experiment to obtain a clear picture with a size of less than 100 nm.

Ans: Size and shape have been calculated by taking 110 particles ranging from the largest to the smallest one, and the resultant mean was found to be 22.08 ± 0.36 nm, which is already less than 100 nm. Also, due to the lack of an instrumental facility, we are unable to repeat the experiment.

7. Introduction, Results and discussion: Antimicrobial studies, such as antibacterial activity, MIC, and MBC authors should incorporate to describe the mechanisms of the antimicrobial action with relevant references to enhance the quality of the content.

Ans: It has been corrected in the revised manuscript as suggested by the reviewers.

Papers that seem to contribute to the manuscript have been cited in the corrected version of the manuscript at appropriate positions.

We tried to incorporate the comments raised in this paper. If further corrections seem to be needed in the manuscript, let me know.

Thanks, and regards

Khaga Raj Sharma, PhD

Associate Professor

Central Department of Chemistry, Tribhuvan University

Nepal (Corresponding author)

---

## [Decision Letter · Decision Letter 1]

9 Oct 2025

Dear Dr. Sharma,

Thank you for submitting your manuscript to PLOS ONE. After careful consideration, we feel that it has merit but does not fully meet PLOS ONE’s publication criteria as it currently stands. Therefore, we invite you to submit a revised version of the manuscript that addresses the points raised during the review process.

We look forward to receiving your revised manuscript.

Kind regards,

Muthugounder Subramanian Shivakumar, Ph.D.

Academic Editor

PLOS ONE

Journal Requirements:

Additional Editor Comments (if provided):

Reviewer 2

Authors Sharma et al. revised the manuscript titled Green Synthesis of Zinc Oxide Nanoparticles Using Catunaregam spinosa (Thunb.) Triveng for Biological Applications in response to the reviewers’ comments. The revised version has improved the flow of information and the overall quality of the content, though it still requires some language refinements before acceptance.

Reviewer 3

The objective of the authors to prepare ZnO NPs using Catunaregam spinosa is fine, but the whole research write-up appears misleading:

Authors must explain why need to choose specific plant part for synthesis. Are there any medicinal properties? If have, is the same compound responsible for reducing Zn ions? How the synchronize for so called biological applications? Whether synthesized material active to absorb UV, authors tested bactericidal properties.

1. Can you rethink on title? What indicates biological applications? Why said green synthesis whether your title written as Catunaregam spinosa extract used synthesis? I mean your title must be revised in terms of scientific accuracy.

2. Rewrite the abstract, introduction, future prospects, and conclusion in the appropriate direction with addressing the problem specific problem.

3. Discuss the novelty of your work and similar studies for similar kinds of work here

4. Several typos and grammatical errors were found throughout the manuscript; revise it appropriately.

5. What would be the significance of your studies? What is the strongest report/point?

Reviewers' comments:

Reviewer's Responses to Questions

**Comments to the Author**

Reviewer #2: All comments have been addressed

Reviewer #3: (No Response)

2. Is the manuscript technically sound, and do the data support the conclusions?

Reviewer #2: Yes

Reviewer #3: Yes

3. Has the statistical analysis been performed appropriately and rigorously?

Reviewer #2: (No Response)

Reviewer #3: Yes

4. Have the authors made all data underlying the findings in their manuscript fully available?

Reviewer #2: Yes

Reviewer #3: Yes

5. Is the manuscript presented in an intelligible fashion and written in standard English?

Reviewer #2: No

Reviewer #3: No

Reviewer #2: Authors Sharma et al. revised the manuscript titled Green Synthesis of Zinc Oxide Nanoparticles Using Catunaregam spinosa (Thunb.) Triveng for Biological Applications in response to the reviewers’ comments. The revised version has improved the flow of information and the overall quality of the content, though it still requires some language refinements before acceptance.

Reviewer #3: The objective of the authors to prepare ZnO NPs using Catunaregam spinosa is fine, but the whole research write-up appears misleading:

Authors must explain why need to choose specific plant part for synthesis. Are there any medicinal properties? If have, is the same compound responsible for reducing Zn ions? How the synchronize for so called biological applications? Whether synthesized material active to absorb UV, authors tested bactericidal properties.

1. Can you rethink on title? What indicates biological applications? Why said green synthesis whether your title written as Catunaregam spinosa extract used synthesis? I mean your title must be revised in terms of scientific accuracy.

2. Rewrite the abstract, introduction, future prospects, and conclusion in the appropriate direction with addressing the problem specifiec problem.

3. Discuss the novelty of your work and similar studies for similar kinds of work here. 4. see some of pubication for further improvement- https://link.springer.com/article/10.1007/s42452-024-06049-z,
https://iopscience.iop.org/article/10.1088/2053-1591/ad6ca9/meta,
https://www.taylorfrancis.com/chapters/edit/10.1201/9781003181224-13/biogenic-non-biogenic-waste-synthesis-nanoparticles-applications-abhishek-kumar-bhardwaj-ram-naraian-shanthy-sundaram-rupali-kaur 5• Several typos and grammatical errors were found throughout the manuscript; revise it appropriately. 6• What would be the significance of your studies? What is the strongest report/point? Highlight it and make a genuine story to represent your work. It looks messy and fragmented.

**Do you want your identity to be public for this peer review?** For information about this choice, including consent withdrawal, please see our Privacy Policy

Reviewer #2: No

Reviewer #3: **Yes: ** Abhishek Kumar Bhardwaj, Assistant Professor, Department of Environmental Science, Amity University Madhya Pradesh, Gwalior MP India.

---

## [Author Response · Author response to Decision Letter 2]

28 Oct 2025

It has been uploaded in the system.

---

## [Decision Letter · Decision Letter 2]

15 Dec 2025

Synthesis of Zinc Oxide Nanoparticles Using Catunaregam spinosa (Thunb.) Triveng for Biologicals Applications

PONE-D-25-09067R2

Dear Dr. Sharma,

We’re pleased to inform you that your manuscript has been judged scientifically suitable for publication and will be formally accepted for publication once it meets all outstanding technical requirements.

Kind regards,

Muthugounder Subramanian Shivakumar, Ph.D.

Academic Editor

PLOS One

Additional Editor Comments (optional):

Accept

Reviewers' comments:

Reviewer's Responses to Questions

**Comments to the Author**

Reviewer #2: All comments have been addressed

Reviewer #3: All comments have been addressed

2. Is the manuscript technically sound, and do the data support the conclusions?

Reviewer #2: Yes

Reviewer #3: Yes

3. Has the statistical analysis been performed appropriately and rigorously?

Reviewer #2: Yes

Reviewer #3: Yes

4. Have the authors made all data underlying the findings in their manuscript fully available?

Reviewer #2: Yes

Reviewer #3: Yes

5. Is the manuscript presented in an intelligible fashion and written in standard English?

Reviewer #2: Yes

Reviewer #3: No

Reviewer #2: Authors Sharma et al. have revised the manuscript titled “Green Synthesis of Zinc Oxide Nanoparticles Using Catunaregam spinosa (Thunb.) Triveng for Biological Applications” in accordance with the reviewers comments and editorial suggestions. The revised version addresses mostly all the comments, and the overall flow and quality of the manuscript have been improved

Reviewer #3: Now, most of the comments are incorporated in this manuscript, and it can be considered for further publication.

**Do you want your identity to be public for this peer review?** For information about this choice, including consent withdrawal, please see our Privacy Policy

Reviewer #2: No

Reviewer #3: **Yes: ** Abhishek Kumar Bhardwaj, Amity University Madhya Pradesh

---

## [Editor Report · Acceptance letter]

PONE-D-25-09067R2

PLOS One

Dear Dr. Sharma,

I'm pleased to inform you that your manuscript has been deemed suitable for publication in PLOS One. Congratulations! Your manuscript is now being handed over to our production team.

Kind regards,

on behalf of

Dr. Muthugounder Subramanian Shivakumar

Academic Editor

PLOS One